# Relationship of Physiographic Position to Physicochemical Characteristics of Soils of the Flooded-Savannah Agroecosystem, Colombia

**Arcesio Salamanca-Carreño [1],*** , **Mauricio Vélez-Terranova [2]**, **Oscar Mauricio Vargas-Corzo [3]**, **Otoniel Pérez-López [4]**, **Andrés Fernando Castillo-Pérez [1]** and **Pere M. Parés-Casanova [5]**

1 Facultad de Medicina Veterinaria y Zootecnia, Universidad Cooperativa de Colombia, Villavicencio 500001, Colombia
2 Facultad de Ciencias Agropecuarias, Universidad Nacional de Colombia, Palmira 763531, Colombia
3 Fedegan-Fondo Nacional del Ganado, Arauca 810001, Colombia
4 Corporación Colombiana de Investigación Agropecuaria, Villavicencio 500001, Colombia
5 Generalitat de Cataluña, 25798 La Seu d'Urgell, Catalonia, Spain
* Correspondence: asaca_65@yahoo.es

**Abstract:** Savannah floodplains are a natural agroecosystem located in the eastern plains of Colombia, with soils considered to be of low fertility. This assumption has not been rigorously validated by direct experimental studies. The aim of the study was to analyze the soils' physicochemical characteristics of the "banks" and "lows", which are physiographic positions, from the floodplain savannah in Arauca, Colombia. Soil samples were collected in "low" ($n = 14$) and "bank" ($n = 15$) physiographic positions. For each soil sample, the following chemical variables were determined: pH, organic carbon (OC), total nitrogen (TN), P, K, Ca, Mg, Na, exchange acidity, cation-exchange capacity (CEC), Fe, Cu, Mn, Zn and B, and physical variable (texture). The Wilcoxon non-parametric test (Mann–Whitney) was applied for the comparison of the soil's physicochemical variables in each physiographic position ($p < 0.05$). The highest values for each variable analyzed correspond to the physiographic position of "low" ($p < 0.05$). The pH, T.N., Na, K and B were not statistically significant ($p > 0.05$). The physiographic positions of "bank" and "low" of floodplain savannah presented low levels of most nutrients, with slightly higher values in the "low" physiographic position. Corrective measures must be applied to improve the nutritional values of savannah soils and, consequently, the productivity of native forages. Despite these deficiencies, the vegetation cover is given by very well-adapted native grasses, reflecting the conditions of said agroecosystem.

**Keywords:** acid soils; clay soils; tropical climate; warm season

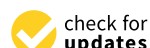



## 1. Introduction

The agroecosystem of the savannah floodplain is located on the left mound of the Meta River, corresponding to the depression of the departments of Arauca and Casanare, in eastern Colombia. It constitutes a characteristic biome, given the climatic seasonality, environmental variability, and availability of water and nutrients [1,2]. The soils are fine-textured, with strong drainage limitations and an impermeable silt-clay-ferruginous substrate [3].

The agroecosystem is made up of different physiographic positions defined by relief and drainage: (a) flat-to-slightly-concave topography in the highest areas, where the "savannah banks" are found, covered by native grasses that constitute the basis of forage feeding for livestock during the rainy season; (b) the low areas, known as "lows", are flooded during the rainy season, and from the forage point of view they support livestock during the dry period [4,5]. The soils are considered acidic and have a medium level of fertility and susceptibility to periodic flooding [6].

Soil physicochemical-characterization studies in the region are scarce, and have not been carried out systematically, considering that they are important to recommend mineral supplementation plans [7]. Calcium values between 114 ppm and 1643 ppm are reported; for magnesium, values from 35 ppm to 358 ppm are considered deficient, while the potassium content is limiting [3], deficiencies that are reflected in the low forage-production and consequently in the production of bovines [8]. Some studies of flooded savannah soils report very low calcium, magnesium, potassium, and other mineral contents, and these may even be absent [9]. Therefore, it is necessary to know the amounts of inorganic nutrients (minerals) that the soil contains, to establish an adequate fertilization for its conservation and sustainability and avoiding its degradation [10].

The analysis of the productive particularities and limitations shown by the soils of the various environments is important to decide their use and establish sustainable activities [11]. Analyzing the quality of the soil of an agroecosystem contributes to the conservation of biodiversity, maintaining environmental quality and promoting the health of plants and animals [12]. Soil analysis is also one of the most important elements in achieving sustainable agriculture [13]. Organic matter and total nitrogen in the soil are essential properties for assessing soil fertility, and constitute transcendental elements for plant nutrition [14]. Soil is a natural component that fulfills functions in the production of food for humans and animals, as well as for maintaining the quality of the environment, so its quality must be assessed [15].

The scarce studies on the floodplain agroecosystem have been based on biotic characterization and description of bodies of water through consultancy by private companies, and are not available to the public, indicating that there is an "information gap" [16,17]. In general, the natural fertility of floodplain savannah soils is reported to be low [2,4,9], although this assumption has not been rigorously validated by direct experimental studies in each of the physiographic positions. Our study hypothesizes that the soils of the two savannah-floodplain physiographic positions may have low nutrient levels. The aim of the study was to analyze the soils' physicochemical characteristics belonging to the "banks" and "lows" physiographic positions in the savannah floodplain in Arauca, Colombia.

## 2. Materials and Methods

### 2.1. Study Site

This study was carried at out the Clarinetero Territorial Division Center, Department of Arauca, eastern Colombia (Figure 1). It is a region of flat topography and savannah floodplain, with physiographic positions of "bank" and "low" (Latitude: 7°08′17″ N, Longitude: 70°59′59″ W). For four months (June to September) the agroclimatic variables reported by a portable weather station were recorded, obtaining values of 646 mm of precipitation, environmental temperature (22.8 °C to 33.0 °C) (Figure 2), relative humidity (90.1%) and 125 m.s.n.m. The study region corresponds to a subhumid tropical-forest zone [18].

The climate and soil situations have special characteristics that influence the management of the agroecosystem, with a period of high rainfall between the months of April to November, during which the savannahs are flooded because of the rains and water coming from the overflow of the main rivers, and a period of drought between the months of December to March, with a water deficit that largely limits agricultural production [5]. The duration of seasonal climatic periods can vary locally [2]. The soils are classified as ultisols and oxisols, and considered of low natural fertility [4,9,19]. Drainage is slow, with the presence of an impermeable horizon [20].

### 2.2. Physiographic Positions

The study was carried out in the following physiographic positions of the savannah floodplain:

Physiographic position of "low". They are areas that are in the basal part of the "bank", limited by fluctuations in the water table, and remain with a sheet of water during the

rainy season (Figure 3). Vegetation cover includes the native grasses *Acroceras zizanioides* (Kunth) Dandy (1931), (blackwater straw); *Hymenachne amplexicaulis* (Rudge) Nees (1829) (water straw); *Leersia hexandra* Sw. (1788) (Lambedora grass); and *Paratheria prostrata* Griseb. (1866) (Carretera grass), among others [5,20]. These pastures constitute the source for feeding bovines in the rainy season. The growth of the grasses is stoloniferous and rhizomatous. The availability of forage is reduced during the dry period because the species are hydrophilic, that is, they depend on humidity in the soil and, in some cases, on the sheet of water [5].

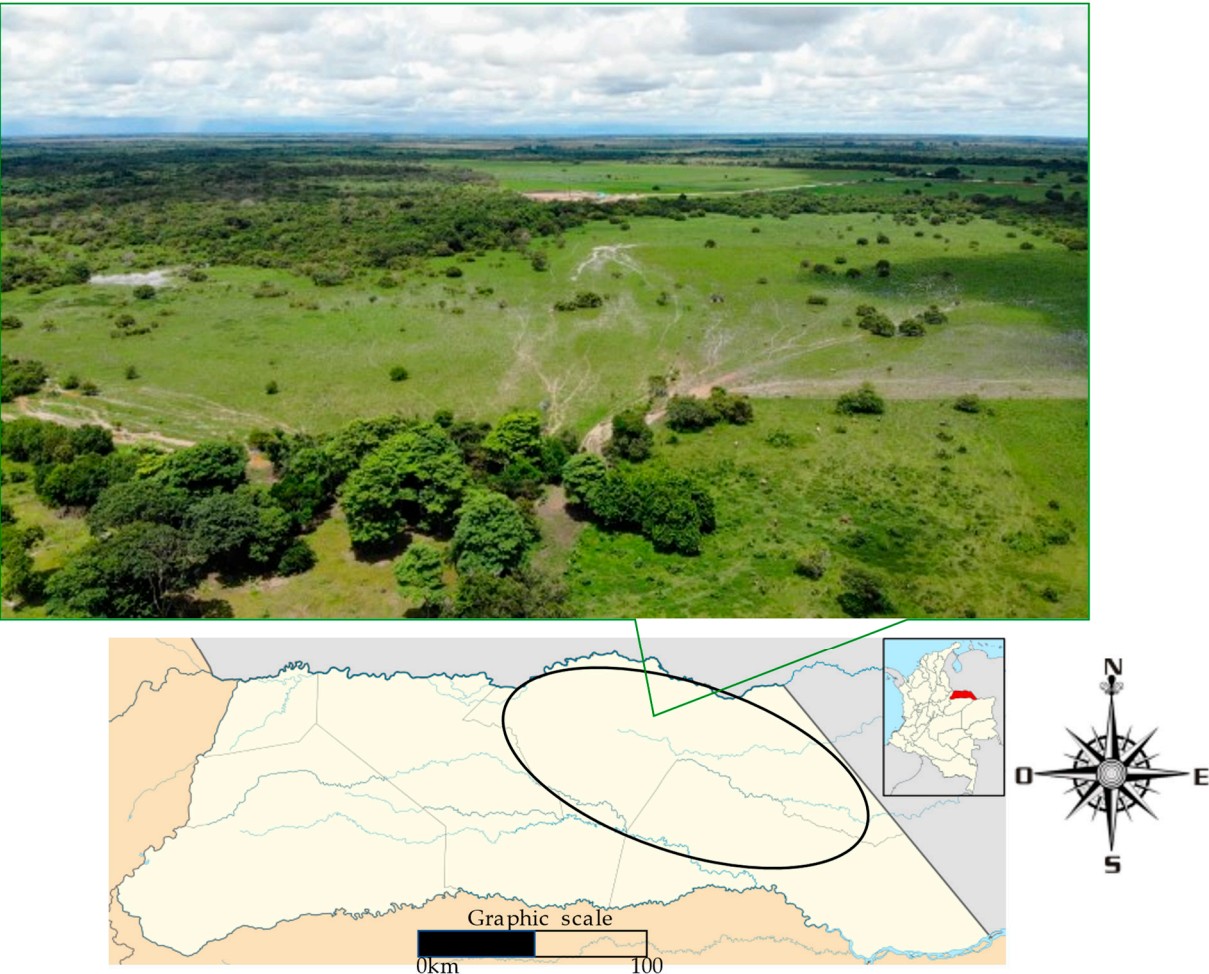

**Figure 1.** Department of Arauca in eastern Colombia. Circle: savannah floodplain agroecosystem, department of Arauca. Red color: location of the department of Arauca, eastern Colombia. Above photograph: savannah floodplain agroecosystem, eastern Colombia (latitude: 07°08′17″ N, longitude: 70°59′59″ W).

Physiographic position of "bank". They are high areas with a convex surface, of variable length and width, parallel to the natural drainages of the flood plain (Figure 3). The vegetation cover includes native grasses like *Paspalum plicatulum* Michx. (1803) (black "bank" grass); *Panicum versicolor* (EPBicknell) Neiuwl. (1911) (white "bank" grass); *Axonopus purpussi* (Mez) Chase (1927) (Guaratara grass); *Axonopus compressus* (Sw) P. Beauv. (1812) (native grass); *Paspalum dilatatum* Poir. (1804); and *Imperata contracta* Hitchc. (1893), among others [5,20]. The growth of the grasses is stoloniferous and rhizomatous. These pastures are the source of food for bovines in the region during the rainy and dry periods. The forage potential of the species has been classified as high, medium, low or none, according to their abundance, frequency, nutritional quality, and animal consumption [5].

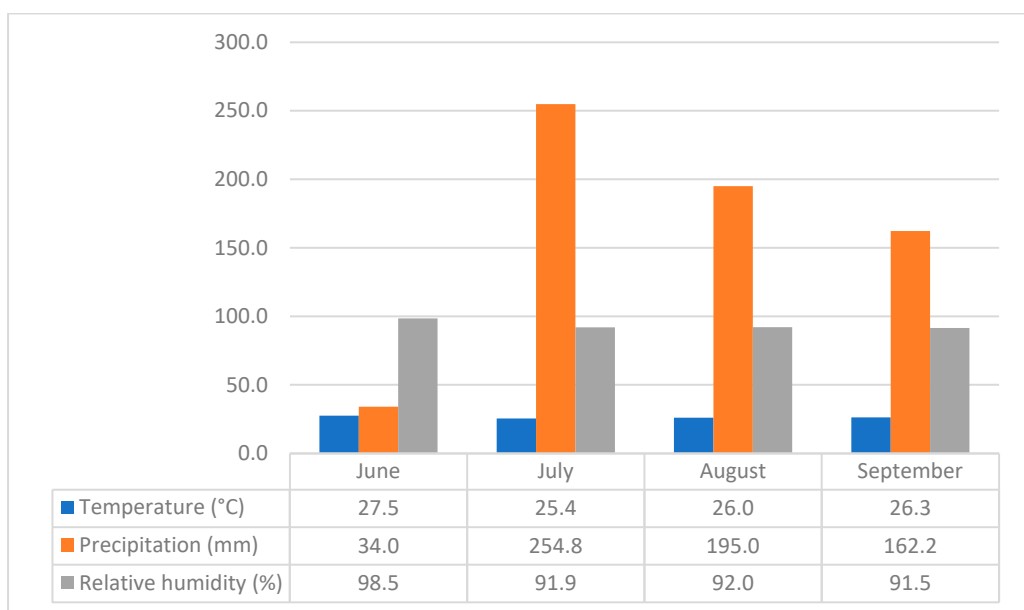

**Figure 2.** Agroclimatic variables of the study site reported by the portable weather station during four months of study.

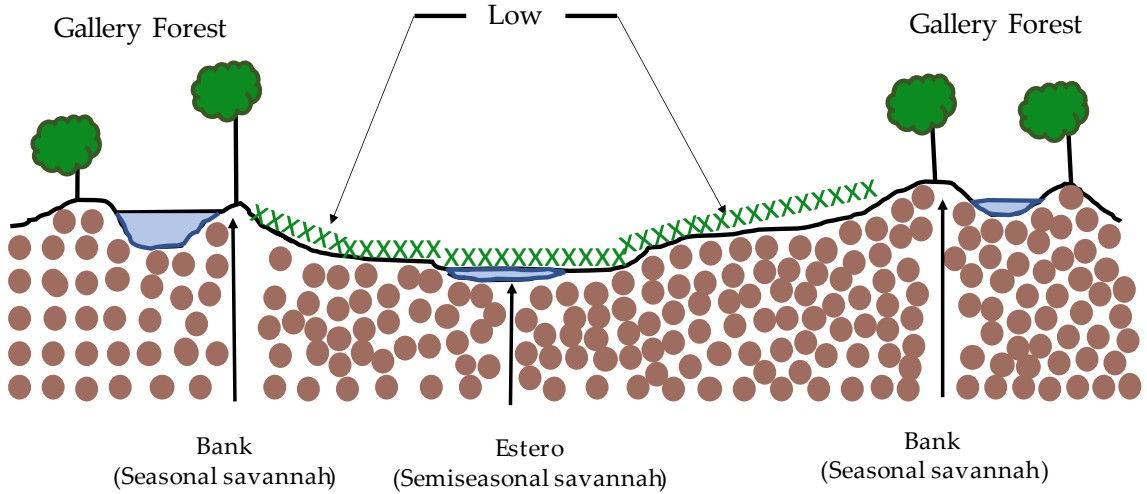

**Figure 3.** Topographic profile of the "bank" and "low" physiographic positions present in the Araucanian floodplain, eastern Colombia (Arauca, Colombia, 2021). Adapted: [5,21].

According to the observation of the authors, the presence of pasture (grasses and legumes) in the paddocks is harmonized with the physiographic position and the climatic period of the year. Some species disappear in the rainy season and others in the dry season, and vice versa. A greater diversity of plant species is observed in "bank" areas compared to "low" areas; In both situations there are limitations due to an excess or deficiency of rainfall. However, most of the native pastures found in the physiographic positions constitute the feeding base for bovines in the region.

*2.3. Soil Samples and Analysis*

A total of twenty-nine soil samples were collected with a shovel, distributed in the physiographic positions of "low" (*n* = 14) and "bank" (*n* = 15). The samples were taken from oxisol soils [9,19]. From 0.80 to 1 kg of soil were extracted at a depth of 30 cm and stored in labeled and sealed plastic bags; subsequently, a pre-drying was carried out in the natural environment until reaching a humidity content between 15 and 24 per cent [22]. The independent samples of each physiographic position ("bank" and "low") were sent to

the laboratory to determine their characteristics, following the procedures established in the Colombian Technical Standards (CTS) of the Colombian Institute of Technical Standards and Certification [23].

Soil samples were taken in areas where a high bovine-grazing frequency were observed. The samples were collected, avoiding an excess of humidity. For each soil sample collected in each physiographic position, the following chemical variables were determined: pH, organic carbon (OC), total nitrogen (TN), P, K, Ca, Mg, Na, exchange acidity, cation-exchange capacity (CEC), Fe, Cu, Mn, Zn and B, and texture. The soil samples were analyzed in the Soil, Water and Foliar Laboratory of the National University of Colombia, Orinoquia headquarters [24]. The analysis methods and the regulations applied for each variable are shown in Table 1.

**Table 1.** Methods of analysis for the determination of physical and chemical variables in physiographic position of the Araucanian flooded savannah (Arauca, Colombia, 2021).

| Trial | Extraction/Digestion Method | Quantification Method | Normative Document |
|---|---|---|---|
| pH determination | Soil: water ratio (*w/v*) 1:1 | Potentiometric | CTS-5264 2008-03-26 |
| Organic Carbon Determination | Walkley–Black Wet Oxidation | Titrimetric | CTS-5403 2013-07-17 |
| Total Nitrogen Determination | Kjendalh digestion | Titrimetric | SOP-S014 |
| Exchangeable bases (Na, Ca, Mg and K) determination | Extraction with ammonium acetate 1N y pH 7 | A.A.S | CTS-5349 2016-09-29 |
| Exchangeable Acidity Determination | Extraction with KCI 1M | Titrimetric | CTS-5263 2017-06-21 |
| Cation-Exchange Capacity Determination | Ammonium-Acetate-Saturation Method 1N y pH 7 | Titrimetric | CTS-5268 2014-01-29 |
| Available Phosphorus (P) Determination | Extraction with Bray II Solution | UV-VIS | CTS-5350 2016-06-15 |
| Available Micronutrients Determination | Extraction with DTPA | A.A.S. | CTS-5526 2007-09-26 |
| Available Boron (B) Determination | Extraction with monobasic Calcium Phosphate | UV-VIS | CTS-5404 2011-07-13 |
| Texture Determination | Dispersion with sodium hexametaphosphate | Bouyoucos/Hydrometer | ACGI-6a edición-2006 |

A.A.S.: atomic absorption spectrophotometry; UV-VIS: ultraviolet visible spectrophotometry; CTS: Colombian Technical Standards; SOP: Standard Operating Procedure; ACGI: Agustin Codazzi Geographical Institute.

### 2.4. Statistical Analysis

The information was analyzed using descriptive statistics (mean values and standard deviation). Initially, the information was reviewed to eliminate outliers. Subsequently, the normality of the information was reviewed through the Shapiro–Wilk test ($p < 0.05$), noting some variables not meeting this assumption. Therefore, the non-parametric Wilcoxon (Mann–Whitney) test was applied for the comparison of the soil physicochemical variables of each physiographic position. For the analysis of the texture variable, frequency tables were obtained in each position. All the data were processed with the statistical software Infostat (2020) free version [25].

### 3. Results

In general, it was observed that the highest values for each variable analyzed corresponded to the physiographic position of "low" ($p < 0.05$) (Table 2). The "lows" presented higher acidity, exchangeable bases (Ca, Mg, K) and CEC, as well as elements such as phosphorus, copper, zinc, and the organic-carbon content. Although no significant differences were found ($p > 0.05$), total nitrogen content was higher in the "low", the Na value was below the quantification limit in both positions, K presented average values slightly

higher in "low", and the amount of B was higher in "bank". With respect to soil texture, a greater amount of silt and clay was observed in the "low", while sand predominated in the "bank" ($p < 0.05$).

**Table 2.** Mean values for the chemical and physical variables according to the physiographic position in the Araucanian flooded savanna (Arauca, Colombia, 2021).

| Variable | Physiographic Position | | W | P-Valor | Reference * | | |
|---|---|---|---|---|---|---|---|
| | "Low" (*n* = 14) | "Bank" (*n* = 15) | | | Tall | Medium | Bass |
| pH | 4.75 ± 0.42 | 5.03 ± 0.46 | 171 | 0.0884 | | | |
| OC (g/kg) | 9.76 ± 1.68 [a] | 7.68 ± 2.0 [b] | 206.5 | 0.0221 | | | |
| TN (g/kg) | 0.78 ± 0.21 | 0.66 ± 0.17 | 137.5 | 0.1125 | >0.20 | 0.10–0.20 | <0.10 |
| Na (meq/100 g) | <0.13 | <0.13 | - | NS | | | |
| K (meq/100 g) | 0.14 ± 0.06 | 0.12 ± 0.06 | 186.5 | 0.1936 | >0.35 | 0.15–0.35 | <0.15 |
| Ca (meq/100 g) | 2.23 ± 0.86 [a] | 1.47 ± 0.56 [b] | 131 | 0.0133 | >6 | 3.00–6.00 | <3 |
| Mg (meq/100 g) | 1.29 ± 0.47 [a] | 0.95 ± 0.35 [b] | 141 | 0.0465 | >2.5 | 1.50–2.50 | <1.50 |
| Al + H (meq/100 g) | 1.88 ± 0.55 [a] | 0.78 ± 0.43 [b] | 60 | 0.0002 | | | |
| CEC (meq/100 g) | 10.88 ± 2.68 [a] | 8.59 ± 2.40 [b] | 227 | 0.029 | | | |
| P (mg/kg) | 9.41 ± 2.28 [a] | 3.86 ± 2.42 [b] | 157 | 0.0009 | >40 | 20–40 | <20 |
| Cu (mg/kg) | 2.45 ± 0.31 [a] | 1.48 ± 0.44 [b] | 254 | 0.0001 | | | |
| Fe (mg/kg) | 307.96 ± 53.39 [a] | 217.3 ± 51.31 [b] | 248.50 | 0.0012 | | | |
| Zn (mg/kg) | 6.60 ± 1.61 [a] | 3.85 ± 1.12 [b] | 100 | 0.0015 | | | |
| Mn (mg/kg) | 16.52 ± 2.89 [a] | 26.28 ± 5.46 [b] | 61 | 0.0004 | | | |
| B (mg/kg) | 0.77 ± 0.27 | 1.02 ± 0.35 | 119 | 0.0732 | | | |
| Silt (%) | 43.12 ± 5.78 [a] | 31.97 ± 5.40 [b] | 250 | 0.0009 | | | |
| Sand (%) | 28.78 ± 5.43 [a] | 51.18 ± 9.63 [b] | 86.50 | 0.0001 | | | |
| Clay (%) | 28.42 ± 4.18 [a] | 13.61 ± 3.64 [b] | 259 | <0.0001 | | | |

* Values included in the results sheet of each analysis. Means with different letters between columns differ statistically ($p < 0.05$). NS: not significant, OC: organic carbon, TN: total nitrogen, CEC: cation-exchange capacity, Al + H: exchange acidity.

With respect to the classification of soil texture, it can be noted that in the physiographic position of "low" the loam or clayey-loam texture predominates (50% and 36% respectively), while in the physiographic unit of "bank", loam or sandy-loam textured soils (53% and 40% respectively) predominate (Table 3).

**Table 3.** Classification of soil texture according to physiographic position ("low" and "bank") of Araucanian flooded-savanna soils (Arauca, Colombia, 2021).

| Variable | "Low" | | "Bank" | |
|---|---|---|---|---|
| | *n* | % | N | % |
| Loam | 7 | 50 | 8 | 53 |
| Clay loam | 5 | 36 | - | - |
| Silt loam | 1 | 7 | - | - |
| Sandy loam | 1 | 7 | 6 | 40 |
| Clayey | - | - | 1 | 7 |

*n* = number of samples.

## 4. Discussion

In this study, the pH (4.75–5.03) of the savannah-floodplain soils is considered strongly acidic, demonstrating a higher acidity in the physiographic position of "low" (4.75) compared to the physiographic position "bank" (5.03) ($p > 0.05$). In general, on the savannah floodplain, acidic soils predominate [9]. The pH value is the result of the material that originates in the soil and the intensity with which the soil-formation processes (resulting from the interaction of climate, a biota made up of vegetation and soil organisms, and relief, over time), acted on it [26]. Due to the climatic and physiographic conditions considered in this study, the decrease in the pH value is possibly related to the constant washing of the

exchangeable bases (Na, K, Ca, and Mg) because of the intense rains, which also affects the increase in the exchange acidity (Al + H) [27,28]. Other studies have found a tendency for soil acidification through the years, and recommend applying correctors such as calcium carbonate [11]

The mean for Al + H was statistically significant ($p < 0.05$), with a higher value for the physiographic position of "low" (1.88 meq/100); however, this result is lower than the value of 3.52 meq/100 reported in soils from the Larense foothills, and is a favorable aspect in terms of soil quality [29]. The presence of Al has a toxic effect on plants, by producing damage to the plasma membrane and affecting cell elongation [30]. In a comparative work [31], they concluded that pH is determinant in the diversity of macrofauna, while copper and phosphorus affect abundance in the agroecosystems analyzed. Organic matter tends to release the hydrogen ion; therefore, soils rich in this component tend to be acidic [26].

The largest amount of organic carbon was observed in the physiographic position of "low" ($p < 0.05$). The organic-carbon values of this study (7.68 to 9.76 g/kg) are within the ranges reported under the Arauca savannah-floodplain conditions, from extremely low (0.09) to high (20.1) [9]. Organic-carbon storage plays a key role in reducing the negative effects of climate change [32]. Likewise, soil organic matter, of which carbon is a part, has nutritive functions for plant growth [32,33]. Soils with little rainfall provide less organic matter [26]. It is also reported that fine-textured soils fix more carbon than soils that have a coarse texture [34]. Therefore, it is considered that the soils in the "low" physiographic position have greater potential for organic-carbon storage. The organic carbon is important, since it provides energy for the heterotrophic organisms that inhabit it [35]. High temperatures and drought can contribute to depressing the concentration of organic matter, since they accelerate the decomposition processes [11]. In this study, it can be revealed that the physiographic position of "low" with respect to the "bank" has a greater capacity for the storage of organic matter, and can also contribute to the carbon cycle of the earth.

Total nitrogen is a fundamental property for evaluating soil fertility [14]. The result for the presence of total nitrogen was non-significant, but slightly higher in the physiographic position of "low" ($p > 0.05$); however, in the two physiographic positions, the total nitrogen content is considered high with respect to the reference values indicated by the laboratory (Table 2). It is recognized that the main source of nitrogen for plants is provided by soil organic matter [36], so it can be deduced from this study that the soils of the physiographic position of "low" provide a greater source of nitrogen to the agroecosystem of the savannah floodplain. When soils have deficit levels of total nitrogen, it is recommended that they are fertilized with nitrogen and conservative management such as direct sowing is performed [11]. Soils with variability and low nitrogen content are used mostly for agricultural purposes [14].

The values of the exchangeable bases (K, Ca, and Mg) were higher in the physiographic position of "low" ($p < 0.05$) (except K), although they were below the reference values. The Na value was reported to be below the quantification limit of the analysis method (<0.13). These results differ from those reported in another study [14] in terms of the mean reference values. The decrease in exchangeable bases occurs due to nutrient leaching because of high rainfall, and this contributes to the increase in exchangeable Al [28]. In general, the analyzed soils are considered to have low K, Ca, and Mg content, which indicates that correctives and fertilizers should be applied to improve forage productivity and, consequently, animal productivity.

The CEC is considered as an indirect indicator of the buffer capacity of the soils, depending on the type and amount of clay [37]. According to the Food and Agriculture Organization of the United Nations, CEC is an indicator of soil fertility, as it represents the soil's ability to retain nutrients [38]. In the present study, the CEC was higher in the "low" than in the "bank" position ($p < 0.05$), possibly associated with the greater presence of clay in this physiographic position, an availability greater than that reported in soils of

Venezuela [39] and similar to the value reported in soils of Honduras [28]. The high values of CEC can also be attributed to the high proportion of organic matter in the soils [40].

The P content was found at a low level (<20), and the higher mean value in this study was for the physiographic position of "low" ($p < 0.05$). P deficiencies are frequent in acid-soil conditions, which limits the production of pastures and is derived from the reactivation of Fe and Al hydroxides with inorganic phosphates, and by constant changes in land use; P is a necessary nutrient for plants and livestock, and its concentrations in the soil can be attributed to its heterogeneity [40,41]. P deficiencies are possibly influenced by the impact of the droughts that affect the region, since moisture is reported to reflect a critical deficit in the soil profile of less than 60 mm of water [11]. Despite the low P concentrations found in the soils, native plants grow and develop in these conditions, considering it as a process of adaptation to these agroecosystems.

The amounts of the microelements Cu, Fe, and Zn were higher in the "low" physiographic position ($p < 0.05$), while the mean value of Mn was higher in the "bank" ($p < 0.05$). In other investigations, higher values for Cu, Zn and Mn were reported, while for Fe the value was lower, and was affected by multiple factors, including the geographical location [42]. The savannah-floodplain soils reflect the conditions of this agroecosystem and, despite the low nutrient content, the availability of native-forage species is perceived.

The amount of B available was not significant ($p > 0.05$) for the two physiographic positions of the floodplain however, a greater availability was found in the physiographic position of "bank". When the availability of B is less than 5 mg kg$^{-1}$, it is considered low [43], an amount that applies to the present study. The B content in the soil varies from 2 mg kg$^{-1}$ to 200 mg kg$^{-1}$ and contributes to the transport of sugars in the cell membranes of plants [44]. In another study, high concentrations of B (221.0 mg kg$^{-1}$ and 346.5 mg kg$^{-1}$) were reported, and these concentrations indicate that this mineral is regulated by factors such as texture, pH, and temperature, among others [45].

With respect to texture, soils with sand content greater than 50% represent limitations for humidity retention, and soils with clay content greater than 50% have an increased susceptibility to waterlogging [28]. Other studies report that most of the carbon is associated with clay and silt particles [46], a condition that for this study could be associated with the highest fertility values in the physiographic position of "low", presenting mean values higher than the physiographic position of "bank".

In general, the soil component in the savannah-floodplain agroecosystem presents contrasting elements, which are more influenced by the alterations produced by the deposits than by the genesis [6,47]. The subsidence presented by the relief of the agroecosystem therefore generates a vast extension with prolonged flooding. The physiography influences in a way that the nutrient flows laterally towards the interior, following the few low slopes that occur [2].

## 5. Conclusions

The "bank" and "low" physiographic positions of the Araucanian savannah floodplain presented low levels of most nutrients, with slightly higher values in the "low" physiographic position. They are acidic soils, with a different texture in the two physiographic positions. Despite these deficiencies, the vegetation cover is given by well-adapted native grasses and legume species, reflecting the conditions of said agroecosystem. Given the small sample size, it is necessary to continue with the study of the Araucanian savannah floodplain soils to investigate more strictly the physical and chemical properties. Corrective measures must be applied to improve the nutritional values of savannah soils and consequently the productivity of native forages. Further studies considering a larger sample size and a complete soil profile (considering B and C horizons) are required, to rigorously investigate the physical and chemical properties of the Araucanian savannah-floodplain soils.

**Author Contributions:** Conceptualization, writing—original draft, and writing—proofreading and editing, A.S.-C. and M.V.-T.; formal analysis, methodology and research, M.V.-T. and A.S.-C.; writing—reviewing and conceptualization, A.S.-C., M.V.-T. and O.P.-L.; conceptualization and research, M.V.-T., O.M.V.-C., A.F.C.-P. and P.M.P.-C. All authors have read and agreed to the published version of the manuscript.

**Funding:** This research was funded by the Research Committee—CONADI (ID2957)—of the Universidad Cooperativa de Colombia.

**Institutional Review Board Statement:** Not applicable, as the study did not involve humans or animals.

**Data Availability Statement:** Data are available upon reasonable request to the second author.

**Acknowledgments:** The authors thank the auxiliary students who collaborated in the collection of field data, as well as the owner of the Villa Cristina farm. All persons included in this section have consented to the acknowledgments.

**Conflicts of Interest:** The authors declare that there is no competing interest regarding the publication of this article.

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
