# Peer review of "Relationship of Physiographic Position to Physicochemical Characteristics of Soils of the Flooded-Savannah Agroecosystem, Colombia"

_agriculture, doi:10.3390/agriculture13010220_

Round 1

Reviewer 1 Report

The manuscript entitled “Relationship of physiographic position with physicochemical characteristics of soils of the flooded savannah agro ecosystem, Colombia” needs a minor revision. In my opinion, the Introduction sections should include the hypothesis of the study and also include some latest references related to this study. Only in this way, the manuscript will be more consistent. In general, the experiment was well performed. However, the discussion part is poor and not well discussed. The conclusions should answer the hypothesis of your study and should focus on the implication of your findings. Some of the comments mentioned itself in the manuscript.   The other comments are given on the manuscript itself.

Author Response

Los autores agradecen los comentarios esclarecedores.

Adjuntamos las correcciones y respuestas.

Respuestas del primer revisor

Comentario

El manuscrito titulado “Relación de la posición fisiográfica con las características fisicoquímicas de los suelos del agroecosistema sabana inundable, Colombia” necesita una revisión menor. En mi opinión, las secciones de Introducción deberían incluir la hipótesis del estudio y también incluir algunas referencias más recientes relacionadas con este estudio. Sólo así el manuscrito será más consistente. En general, el experimento fue bien realizado. Sin embargo, la parte de discusión es pobre y no está bien discutida. Las conclusiones deben responder a la hipótesis de su estudio y deben centrarse en la implicación de sus hallazgos. Algunos de los comentarios se mencionan a sí mismos en el manuscrito. Los demás comentarios se dan en el propio manuscrito.

Respuesta

La hipótesis está incluida en el texto.

No existen referencias recientes ya que los estudios de suelos en el agroecosistema sabana inundable araucana son casos. Se considera que este es un primer estudio sobre el análisis de las características físico-químicas de los suelos del agroecosistema sabana inundable.

Expande la discusión en el texto

Se corrige la discusión

Todas las correcciones incluidas en el texto.

Inglés corregido

Reviewer 2 Report

Review report

1.     Manuscript Title

Relationship of physiographic position with physicochemical characteristics of soils of the flooded savannah agroecosystem, Colombia

The Title of the review paper is well formulated.

2.     Abstract

The rationale of the study is lacking at the opening of the abstract

Lines 14 & 15. Flooded savannahs are a natural agroecosystem located in the eastern plains of Colombia, with soils considered to be of low fertility. What is the problem related with this fact that needs to be addressed through the study?

Line 15.   ……the objective was too…… The objective of what? Do you mean the objective of the research or the study or what?

Line 24. Physiographic position of "low"

What does the term “physiographic position” of “low” mean?

Lines 27 & 28. The physiographic positions of "bank" and "low" of floodable savannah presented low levels of most nutrients, with slightly higher values in the "low" physiographic position

What do the terms “bank”, “low”, etc. mean?

Where is the conclusion of the abstract? What were they key results?

What are the implication of the results?

3.     Introduction

·       The problem formulation is vague.

·       The consequences of maintaining the status quo is not clearly indicated. That means, what would or could happen if the indicated soil problems are not addressed.

·       The research questions are not clear!

4.     Materials and Methods

Okay

5.     Results

Okay, but the physiological characteristics “low”, “bank”, etc. are not defined clearly.

6.     Discussions

Okay, but improve the language.

Conclusion

The conclusion is not really conclusive as such.

Please, indicate the key findings, interpret them; write the implications of the results and outlook of future research.

7.    References

Okay

The manuscript is accepted with minor modification.

Author Response

Respuestas del segundo revisor

Los autores agradecen los comentarios esclarecedores.

Adjuntamos las correcciones y respuestas.

Comentario

  1. Resumen

Falta la justificación del estudio en la apertura del resumen.

Líneas 14 y 15.  Las sabanas inundables son un agroecosistema natural ubicado en los llanos orientales de Colombia, con suelos considerados de baja fertilidad. ¿Cuál es el problema relacionado con este hecho que necesita ser abordado a través del estudio?

Respuesta

Corregido el resumen en el texto.

Comentario

línea 15 . ……el objetivo era también…… ¿El objetivo de qué? ¿Se refiere al objetivo de la investigación o del estudio o qué?

Respuesta

In the abstract the objective of the study was corrected: “The aim of the study was to analyze the soils physicochemical characteristics belonging to the “banks” and “lows” physiographic positions from the floodplain savannah in Arauca, Colombia”

Comment

Line 24. Physiographic position of "low"

What does the term “physiographic position” of “low” mean?

Response

The following paragraph is written in the introduction:

The agroecosystem is made up of different physiographic positions defined by relief and drainage: a) flat to slightly concave topography are the highest areas where the "savannah banks" are found covered by native grasses that constitute the basis of forage feeding for livestock during the rainy season; b) the low areas, known as "lows", are flooded during the rainy season and from the forage point of view they support livestock during the dry period [4,5].

Similarly, materials and methods (2.1. Study site) describe the physiographic position of low:

Physiographic position of “low”. They are areas that are in the basal part of the "bank", limited by fluctuations in the water table and remain with a sheet of water during the rainy season (Figure 3b). Vegetation cover includes native grass Acroceras zizanioides (Kunth) Dandy (1931), (black water straw); Hymenachne amplexicaulis (Rudge) Nees (1829) (water straw); Leersia hexandra Sw. (1788) (Lambedora grass); Paratheria prostrata Griseb. (1866) (Carretera grass), among others [5,19].

Comment

Lines 27 & 28. The physiographic positions of "bank" and "low" of floodable savannah presented low levels of most nutrients, with slightly higher values in the "low" physiographic position

What do the terms “bank”, “low”, etc. mean?

Response

The following paragraph is written in the introduction:

“The agroecosystem is made up of different physiographic positions defined by relief and drainage: a) flat to slightly concave topography are the highest areas where the "savannah banks" are found covered by native grasses that constitute the basis of forage feeding for livestock during the rainy season; b) the low areas, known as "lows", are flooded during the rainy season and from the forage point of view they support livestock during the dry period [4,5].”

Similarly, materials and methods (2.1. Study site) describe the two physiographic positions. An image is included

Physiographic position of "bank". They are high areas with a convex surface, of variable length and width, parallel to the natural drainages of the flood plain (Figure 3a). The vegetation cover includes native grass Paspalum plicatulum Michx. (1803) (black “bank” grass); Panicum versicolor (EPBicknell) Neiuwl. (1911) (white “bank” grass); Axonopus purpussi (Mez) Chase (1927) (Guaratara grass); Axonopus compressus (Sw) P. Beauv. (1812) (native grass); Paspalum dilatatum Poir. (1804), Imperata contracta Hitchc. (1893), among others [5,19].

Physiographic position of “low”. They are areas that are in the basal part of the "bank", limited by fluctuations in the water table and remain with a sheet of water during the rainy season (Figure 3b). Vegetation cover includes native grass Acroceras zizanioides (Kunth) Dandy (1931), (black water straw); Hymenachne amplexicaulis (Rudge) Nees (1829) (water straw); Leersia hexandra Sw. (1788) (Lambedora grass); Paratheria prostrata Griseb. (1866) (Carretera grass), among others [5,19].

Figure 3. Physiographic positions topographical profile present in the Araucanian flooded savannah, eastern Colombia. (a) Physiographic position “bank”; (b) Physiographic position “low” (Arauca, Colombia, 2021). Adapted: [5]

Comment

Where is the conclusion of the abstract? What were they key results?

Response

In the summary are the conclusions and the results

What are the implication of the results?

Response

Corrected in text. In the Conclusions item are the implications

Comment

  1. Introduction
  • The problem formulation is vague.
  • The consequences of maintaining the status quois not clearly indicated. That means, what would or could happen if the indicated soil problems are not addressed.
  • The research questions are not clear!

Response

Suggested fixes are included in the introduction.

  1. Materials and Methods

Okay

 Comment

  1. Results

Okay, but the physiological characteristics “low”, “bank”, etc. are not defined clearly.

Response

The following paragraph is written in the introduction:

The agroecosystem is made up of different physiographic positions defined by relief and drainage: a) flat to slightly concave topography are the highest areas where the "savannah banks" are found covered by native grasses that constitute the basis of forage feeding for livestock during the rainy season; b) the low areas, known as "lows", are flooded during the rainy season and from the forage point of view they support livestock during the dry period [4,5].

Similarly, materials and methods (2.1. Study site) describe the two physiographic positions. An image is included

Physiographic position of "bank". They are high areas with a convex surface, of variable length and width, parallel to the natural drainages of the flood plain (Figure 3a). The vegetation cover includes native grass Paspalum plicatulum Michx. (1803) (black “bank” grass); Panicum versicolor (EPBicknell) Neiuwl. (1911) (white “bank” grass); Axonopus purpussi (Mez) Chase (1927) (Guaratara grass); Axonopus compressus (Sw) P. Beauv. (1812) (native grass); Paspalum dilatatum Poir. (1804), Imperata contracta Hitchc. (1893), among others [5,19].

Physiographic position of “low”. They are areas that are in the basal part of the "bank", limited by fluctuations in the water table and remain with a sheet of water during the rainy season (Figure 3b). Vegetation cover includes native grass Acroceras zizanioides (Kunth) Dandy (1931), (black water straw); Hymenachne amplexicaulis (Rudge) Nees (1829) (water straw); Leersia hexandra Sw. (1788) (Lambedora grass); Paratheria prostrata Griseb. (1866) (Carretera grass), among others [5,19].

Figure 3. Physiographic positions topographical profile present in the Araucanian flooded savannah, eastern Colombia. (a) Physiographic position “bank”; (b) Physiographic position “low” (Arauca, Colombia, 2021). Adapted: [5]

Comment

  1. Discusiones

Está bien, pero mejora el lenguaje.

Respuesta

Se corrige y amplía la discusión. se incluyó en el texto

 Comentario

Conclusión

La conclusión no es realmente concluyente como tal.

Por favor, indique los hallazgos clave, interprételos; escribir las implicaciones de los resultados y las perspectivas de futuras investigaciones.

Respuesta

Las conclusiones fueron corregidas de acuerdo a las sugerencias. Se escriben algunas implicaciones y se recomiendan estudios futuros.

  1. Referencias

Bueno

El manuscrito se acepta con modificaciones menores .

Inglés corregido

Reviewer 3 Report

Improve the cartographic design of the location map of the study area.

A figure with a topographic profile would be helpful to understand the differences between sites better.

Improve the environmental information of the study site; include a climogram per study site; include photographs of soil profiles and, if possible, laboratory data or at least field descriptions; include the types of rocks in both topographic positions, etc. A better and more explicit description of the native and cultivated grasses, as well as a figure, shows the roots' depth and extent.

It is advisable to cite the original sources of soil analysis techniques rather than the local standard.

Explore other types of multivariable data analysis, for example, principal component analysis, discriminant analysis, or others.

It would be ideal for comparing the total contents of the elements along the profile expressed in quantity per unit area, for example, in kg/m2 or ton/ha. Please you see Soil & Environment software.

In the text and table 3, you must replace Frank with Loam

Do not use abbreviations for organic carbon and total nitrogen

CEC instead of C.E.C.

It is recommended that the authors mention the weak parts of the work, such as the lack of data from complete soil profiles, because the quality of the subsoil is unknown (horizons B and C).

This study reported information only on topsoil (30 cm depth). We only have a general idea of what happens in less than a third of the soil. 

Author Response

Los autores agradecen los comentarios esclarecedores.

Adjuntamos las correcciones y respuestas.

Respuestas del tercer revisor

Comentario

Mejorar el diseño cartográfico del mapa de ubicación del área de estudio.

Respuesta

Se mejoró el mapa de ubicación del sitio de estudio (Figura 1).

Comentario

Una figura con un perfil topográfico sería útil para comprender mejor las diferencias entre los sitios.

Respuesta

Se incluye en el texto una figura con un perfil topográfico (Figura 3)

Comentario

Mejorar la información ambiental del sitio de estudio; incluir un climograma por sitio de estudio; incluir fotografías de perfiles de suelo y, si es posible, datos de laboratorio o al menos descripciones de campo; incluir los tipos de rocas en ambas posiciones topográficas, etc. Una mejor y más explícita descripción de los pastos nativos y cultivados, así como una figura, muestra la profundidad y extensión de las raíces.

Respuesta

Item 2.1 describes the environmental information of the study site

The climogram of the study site is included (Figure 2).

Native grasses are described

Soil samples were only taken to study the physicochemical characteristics of the two physiographic positions. It was not object to study the extension or depth of the roots. It will be considered for future studies. Soil profiles were not analyzed either.

The soil samples were analyzed in the National University Laboratory (Bibliography 22) following the procedures established in the Colombian Technical Standards (CTS) of the Colombian Institute of Technical Standards and Certification (Bibliography 21). The information is available in table 1.

Comment

It is advisable to cite the original sources of soil analysis techniques rather than the local standard.

Response

Table 1 shows the methods used to determine the physicochemical variables of the soil. There is also the Colombian regulation for said analyzes

For the analysis was utilization in the Colombian Technical Standards (CTS) of the Colombian Institute of Technical Standards and Certification (Bibliography 21).

Comment

Explore other types of multivariable data analysis, for example, principal component analysis, discriminant analysis, or others.

Response

Since the document presented is a communication, we prefer to focus on a direct comparison between the physiographic units using the Wilcoxon non-parametric test. It is important to mention that another paper is currently being developed where the quality of soils is related to the nutritional quality of native forages and the climate of the area, using in this case multivariate analysis methods as principal components, canonical correlations, and partial least squares regressions.

Comment

It would be ideal for comparing the total contents of the elements along the profile expressed in quantity per unit area, for example, in kg/m2 or ton/ha. Please you see Soil & Environment software.

Response

It was not the objective of the study. The objective was to analyze the relationship of the physicochemical characteristics of the Araucanian floodplain savannah soils according to the physiographic position of "low" and "bank".

Comment

In the text and table 3, you must replace Frank with Loam

Response

Corrected in the text

Comment

Do not use abbreviations for organic carbon and total nitrogen

Response

Corrected in the text

Comment

CEC instead of C.E.C.

Response

Corrected in the text

Comment

It is recommended that the authors mention the weak parts of the work, such as the lack of data from complete soil profiles, because the quality of the subsoil is unknown (horizons B and C).

Response

Corrected in the conclusion section

Your recommendation will be considered for future research.

Comment

This study reported information only on topsoil (30 cm depth). We only have a general idea of what happens in less than a third of the soil. 

Response

El objetivo fue comparar las características fisiográfico-químicas de la posición fisiográfica de “banco” y “bajo”. En futuros estudios se analizarán las capas profundas del suelo.

Inglés corregido

Round 2

Reviewer 3 Report

I have observed an improvement in the text, however, there is still much to improve.

This "communication" type article is interesting but since a case of a very particular site is reported, it is necessary to clearly specify the particularities of the site, but unfortunately they still do not achieve the desired quality. Figures 1 (location of the study area with cartographic deficiencies), Figure 2 (incomplete climogram) and figure 3 (deficient topographic profile)

These observations were partially addressed: Improve the cartographic design of the location map of the study area; a figure with a topographic profile would be useful to better understand the differences between sites; and a better and clearer description of the native and cultivated grasses, as well as a figure showing the depth and extent of the roots.

The statistical analysis of the data could be improved with a multivariable analysis.

Use Loam instead of Frank in table 3

It is essential to include the names of the soils using the Soil taxonomy or the WRB. Was the work done in an Ultisol or an Oxisol? As this work is about the differences in the soils, it is necessary to say which soils were worked on.

The document needs an English revision, please see conclusions.

Author Response

Los autores agradecen los comentarios esclarecedores.

Adjuntamos las correcciones y respuestas.

Respuestas del tercer revisor Round2

Comentario

This "communication" type article is interesting but since a case of a very particular site is reported, it is necessary to clearly specify the particularities of the site, but unfortunately they still do not achieve the desired quality. Figures 1 (location of the study area with cartographic deficiencies), Figure 2 (incomplete climogram) and figure 3 (deficient topographic profile)

These observations were partially addressed: Improve the cartographic design of the location map of the study area; a figure with a topographic profile would be useful to better understand the differences between sites; and a better and clearer description of the native and cultivated grasses, as well as a figure showing the depth and extent of the roots.

Response

The description of the study site location has been improved. In Materials and Methods (Item 2.1 Study site) was added: “This study was carried at the Clarinetero Territorial Division Center, Department of Arauca, eastern Colombia”.

 Corrections were included in the text and in Figure 1. Figure 1 explains the circle, the red color, and the above photograph

Figure 1. Department of Arauca in Colombia eastern. Circle: Floodplain savannah agroecosystem department of Arauca. Red color: Location of the department of Arauca, eastern Colombia. Above photograph: Floodplain savannah agroecosystem, eastern Colombia (latitude: 07° 08′ 17″ N, longitude: 70° 59′ 59″ W)

The climogram (Figure 2) was completed

Corrections are included in the text

Figure 2. Agroclimatic variables of the study site reported by the portable weather station during four months of study

The topographic profile has been improved (Figure 3). In item 2.2 (Physiographic positions) a description of the native grasses was made.

Corrections are included in the text

Figure 3. Topographic profile of the “bank” and “low” physiographic positions present in the Araucanian floodplain, eastern Colombia (Arauca, Colombia, 2021). Adapted: [Bibliography 5 and 21]

Physiographic position of “low”. They are areas that are in the basal part of the "bank", limited by fluctuations in the water table and remain with a sheet of water during the rainy season (Figure 3). Vegetation cover includes native grass Acroceras zizanioides (Kunth) Dandy (1931), (black water straw); Hymenachne amplexicaulis (Rudge) Nees (1829) (water straw); Leersia hexandra Sw. (1788) (Lambedora grass); Paratheria prostrata Griseb. (1866) (Carretera grass), among others [Bibliography 5 and 20]. These pastures constitute the source for feeding bovines in the rainy season. The availability of forage is reduced during the dry period because the species are hydrophilic, that is, they depend on humidity in the soil and, in some cases, on sheet of water [Bibliography 5].

Physiographic position of "bank". They are high areas with a convex surface, of variable length and width, parallel to the natural drainages of the flood plain (Figure 3). The vegetation cover includes native grass Paspalum plicatulum Michx. (1803) (black “bank” grass); Panicum versicolor (EPBicknell) Neiuwl. (1911) (white “bank” grass); Axonopus purpussi (Mez) Chase (1927) (Guaratara grass); Axonopus compressus (Sw) P. Beauv. (1812) (native grass); Paspalum dilatatum Poir. (1804), Imperata contracta Hitchc. (1893), among others [Bibliography 5 and 20]. These pastures are the source of food for bovines in the region during the rainy and dry periods. The forage potential of the species has been classified as high, medium, low or none, according to their abundance, frequency, nutritional quality, and animals consumption [Bibliography 5].

The growth of the grasses (“bank” and low”) is stoloniferous and rhizomatous, and some grasses disappear in the rainy season and others in the dry season.

Corrections are included in the text

Comment

“… as well as a figure showing the depth and extent of the roots”.

Response

It was not the aim of this study to analyze the depth or extension of the roots. The aim of the study was: The aim of the study was to analyze the soils physicochemical characteristics belonging to the “banks” and “lows” physiographic positions from the floodplain savannah in Arauca, Colombia”.

 Comment

The statistical analysis of the data could be improved with a multivariable analysis.

Response

Since the document presented is a communication, we prefer to focus on a direct comparison between the physiographic units using the Wilcoxon non-parametric test. It is important to mention that another paper is currently being developed where the quality of soils is related to the nutritional quality of native forages and the climate of the area, using in this case multivariate analysis methods as principal components, canonical correlations, and partial least squares regressions.

 Comment

Use Loam instead of Frank in table 3

Response

Corrected

Comment

It is essential to include the names of the soils using the Soil taxonomy or the WRB. Was the work done in an Ultisol or an Oxisol? As this work is about the differences in the soils, it is necessary to say which soils were worked on.

Response

En Materiales y Métodos (Ítem 2.3. Sitio de estudio ) agregó: “Las muestras fueron tomadas de suelos Oxisoles [Bibliografía 9 y 19].

El nombre del suelo se incluyó usando la Taxonomía de Suelos [Bibliografía 19]

Comentario

El documento necesita una revisión en inglés, ver conclusiones.

Respuesta

el ingles fue revisado

Las conclusiones fueron revisadas e incluidas en el texto.
